# Descemet Stripping Automated Endothelial Keratoplasty in Thick Corneas

**DOI:** 10.3390/jcm11195601

**Published:** 2022-09-23

**Authors:** Chendi Li, Wenyu Wu, Gege Xiao, Vishal Jhanji, Jing Hong, Yun Feng

**Affiliations:** 1Department of Ophthalmology, Peking University Third Hospital, Beijing 100191, China; 2Beijing Key Laboratory of Restoration of Damaged Ocular Nerve, Peking University Third Hospital, Beijing 100191, China; 3Department of Ophthalmology, University of Pittsburgh School of Medicine, Pittsburgh, PA 15213, USA

**Keywords:** DSAEK, endothelial cell destiny, corneal thickness, retrospective

## Abstract

Purpose: To evaluate the outcomes of Descemet’s Stripping Automated Endothelial Keratoplasty (DSAEK) in corneas > 820 microns in thickness. Methods: This retrospective study included 30 eyes of 30 patients who underwent DSAEK. Endothelial cell destiny (ECD) and corneal thickness were recorded before surgery and at 1 and 12 months postoperatively. Patients were divided into two groups (≤ 820 microns and > 820 microns) based on median preoperative corneal thickness. Linear regression analyses were used to investigate the correlations between ECD and preoperative corneal thickness. Results: Recipient corneal thickness (RCT) and postoperative central cornea thickness had a statistically significant difference 1 month after surgery (*p* = 0.03, *p* = 0.08, respectively). BCVA and ECD did not have a statistical difference in the two groups at 1 month and 12 months after DSAEK. Conclusions: BCVA, ECD and corneal thickness were similar at 12 months after DSAEK in thick corneas. DSAEK is a viable surgical option in thick corneas.

## 1. Introduction

Endothelial Keratoplasty (EK) is the surgical treatment of choice in patients with endothelial dysfunction and clear corneal stroma. Significant improvements in surgical techniques of EK over the last few years have helped surgeons achieve optimal visual outcomes with minimal complications in the postoperative period [1,2]. Unlike the developed world, where corneal tissue is readily available, the wait for surgery can be a few months in the developing world. Consequently, chronic corneal endothelium dysfunction can lead to severe corneal edema, corneal stromal haze and corneal scarring. Nonetheless, EK is a viable surgical option for these patients. The reported mean endothelial cell (EC) loss after DSAEK varies from 34% to 48% 1 year after surgery [3,4,5,6,7,8]. However, the range of preoperative central corneal thickness was 635–641 microns in these studies [7,8]. There is a paucity of data on surgical outcomes and EC loss in patients with thick corneas preoperatively. In this study, we retrospectively analyzed the outcomes of DSAEK in patients with thick corneas preoperatively.

## 2. Materials and Methods

### 2.1. Patients Selection

Patients who underwent DSAEK between January 2017 and December 2018 at Peking University Third Hospital Eye Institute were enrolled in this study. Inclusion criteria included patients with corneal endothelial dysfunction due to pseudophakic bullous keratopathy (PBK), Fuchs’endothelial corneal dystrophy, blunt ocular trauma and Congenital Hereditary Endothelial Dystrophy (CHED). Exclusion criteria were previous corneal transplantation, glaucoma drainage valve implantation and active ocular infection or inflammation. The study protocol was approved by the ethics committee of the hospital and followed the tenets of the Declaration of Helsinki.

### 2.2. Donor Tissues

The corneoscleral buttons were stored in Optisol -GS medium (Bausch & Lomb, Irvine, CA, USA) at 4 °C. The endothelial cell density (ECD) was quantified using EB-3000 XYZ Eyebank specular microscope (HAI Laboratories Inc., Lexington, MA, USA). All corneas with ECD ≥ 2800 cells/mm^2^ were utilized for surgery.

### 2.3. Surgical Procedure

All surgeries were performed under retrobulbar anesthesia or general anesthesia by two corneal specialists. Donor lenticules were prepared using the Moria ALTK system (Moria SA, Antony, France) in the operating room. DSAEK lenticules were prepared using 7–8 mm donor cornea punches. The donor lenticule was inserted into the anterior chamber using Busin glide (Moria, Doylestown, PA, USA). The postoperative treatment included topical ofloxacin 0.3% eye drops 4 times daily for 1 month. Topical corticosteroid eye drops were used in the form of prednisolone acetate 1% eye drops 4 times daily.

The clinical data were collected preoperatively and at 1 and 12 months postoperatively. The preoperative corneal thickness, recipient corneal thickness and graft thickness were assessed with anterior segment optical coherence tomography (AS-OCT; Carl Zeiss Meditec, Dublin, CA, USA). ECD was measured using laser confocal scanning microscopy (HRT-3: Heidelberg Engineering, Heidelberg, Germany).

### 2.4. Statistical Analysis

All statistical analyses were carried out using SPSS software V 22.0 (SPSS, Inc., Chicago, IL, USA). Statistical analyses were performed using frequency tables, chi-square test and Student *t*-test. A *p*-value < 0.05 was considered statistically significant.

## 3. Results

A total of 30 eyes of 30 patients were included in this study. Patients were divided into two groups based on the median preoperative corneal thickness of 820 microns, into group 1 (corneal thickness ≤ 820 microns) and group 2 (corneal thickness > 820 microns). The baseline patient and donor characteristics are shown in Table 1. Preoperative corneal thickness was significantly different (*p* < 0.001) between the two groups. Gender, age, eye, preoperative BCVA (Best corrected visual acuity) and preoperative diagnoses were comparable between the two groups (*p* > 0.05).

BCVA and ECD were comparable at 1 month and 12 months after DSAEK (*p* > 0.05) (Table 2). Recipient corneal thickness and postoperative central cornea thickness were thinner in group 1 than group 2 at 1 month after surgery (*p* < 0.05), but there was no significant difference at 12 months. Graft thickness was not significantly different between both groups at 1 month and 12 months postoperatively (*p* > 0.05).

Figure 1 and Figure 2 show clinical photographs and anterior-segment OCT before surgery at 1 and 12 months after DSAEK.

Figure 3 and Figure 4 show that postoperative ECD is not associated with preoperative cornea thickness at 1 month (r = 0.03, *p* = 0.88) and 12 months (r = −0.22, *p* = 0.25) after surgery.

We further analyzed cases with and without bullous keratopathy. In cases with bullous keratopathy, compared to preoperative corneal thickness (795.2 ± 108.9 microns), there was a significant change at 1 month (650.1 ± 82.8 microns) and 12 months (641.2 ± 62.9 microns). In non-bullous keratopathy cases, the preoperative corneal thickness (863.3 ± 170.8 microns) did not change significantly at 1 month (762.3 ± 167.6 microns) but the thickness at 12 months (648.3 ± 71.1 microns) was significantly less compared to the preoperative level.

## 4. Discussion

Endothelial keratoplasty is a well-established surgical technique for selective corneal transplantation. The visual outcomes are better in cases with early corneal endothelial dysfunction compared to cases with chronic endothelial damage. Outcomes can be compromised in cases with long-standing endothelial damage mainly due to increased corneal thickness and corneal haze. It is noteworthy that EK can be performed in long-standing bullous keratopathy to alleviate pain and ocular discomfort without the inherent complications associated with penetrating keratoplasty.

Although group 2 had thick preoperative corneas, BCVA was comparable with group 1 at the time of preoperative, 1 month and 12 months after DSAEK. The final mean logMAR BCVA in our study was not as good as the previous report, which ranged from 0.13 to 0.2 [9]. It is due to the low preoperative BCVA and poor corneal condition.

There is no data on the loss of endothelial cells in thick corneas after EK. In the current study, we divided cases based on preoperative corneal thickness (corneal thickness ≤ 820 microns and > 820 microns). We found that endothelial cell density was similar in both groups at 1 and 12 months after DSAEK. Overall, the endothelial cell loss in our study was higher than previous reports, which have reported a cell loss ranging from 34–48% at 1 year after DSAEK [3,4,5,6,7,8]. In our study, most of the EC loss occurred during the first month after surgery, which may be attributed to intraoperative iatrogenic cell loss in the early postoperative period [10].

Although preoperative recipient corneal thickness did not influence ECD 1 year after DSAEK, it was noted that the group with thicker recipient corneas had a higher EC loss between 1 and 12 months compared to the group with thinner recipient corneal thickness. We were unable to explain this difference between the two groups. The corneal thickness decreased to near normal levels within one month of DSAEK surgery in both groups in our study and continued to remain stable up to 1 year after surgery. The recipient corneal thickness was slightly higher at one month postoperatively in thicker corneas at baseline. This is expected since the newly transplanted endothelium would need a longer time to work on a thick cornea compared to a thin recipient cornea.

The main limitations of the current study are the small sample size and lack of a comparative surgical group. We did not have data from any additional follow-up visits. It was noteworthy that the graft thickness did not change much over the 12-month period. We hypothesize that it was likely due to the relative dehydration associated with dextran in the transport medium. Additionally, long-term follow-up beyond 1-year postoperatively will help us to understand if grafts behave differently in thicker corneas. Descemet’s membrane endothelial keratoplasty (DMEK) has been introduced as an alternative to DSAEK for quicker visual recovery [11,12]. It would be interesting to compare DMEK and DSAEK in extremely thick recipient corneas in future studies. In our study, the choice of DSAEK over DMEK was made mainly due to suboptimal visualization intraoperatively. In conclusion, our study showed that ECD and postoperative central cornea thickness 1 year after DSAEK is not correlated with preoperative corneal thickness. The study supports that DSAEK is a viable surgical option in patients with severe edema.

## Figures and Tables

**Figure 1 jcm-11-05601-f001:**
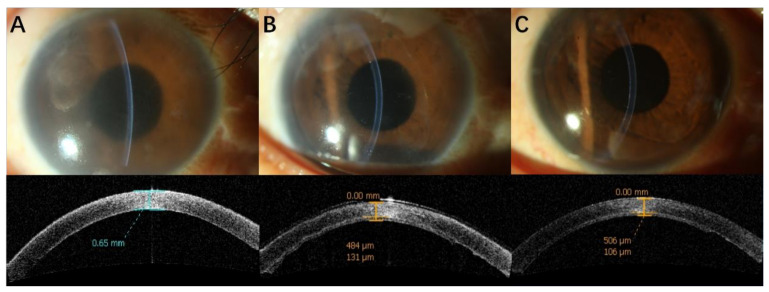
Clinical photographs and AS-OCT of a patient in group 1 before DSAEK (**A**) (CCT650 μm), 1 month after DSAEK (**B**) (RCT484 μm, GT131 μm) and 12 months after DSAEK (**C**) (RCT506 μm, GT106 μm).

**Figure 2 jcm-11-05601-f002:**
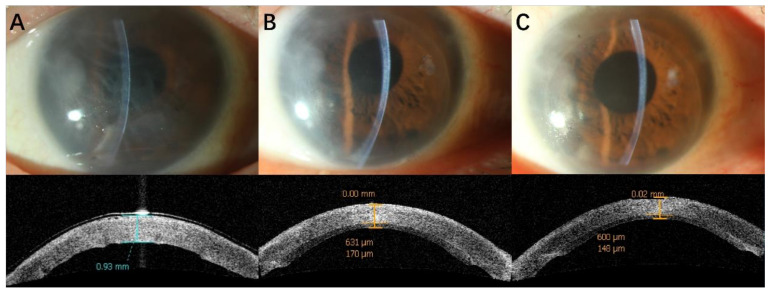
Clinical photographs and AS-OCT of a patient in group 2 before DSAEK (**A**) (CCT930 μm), 1 month after DSAEK (**B**) (RCT631 μm, GT170 μm) and 12 months after DSAEK (**C**) (RCT600 μm, GT148 μm).

**Figure 3 jcm-11-05601-f003:**
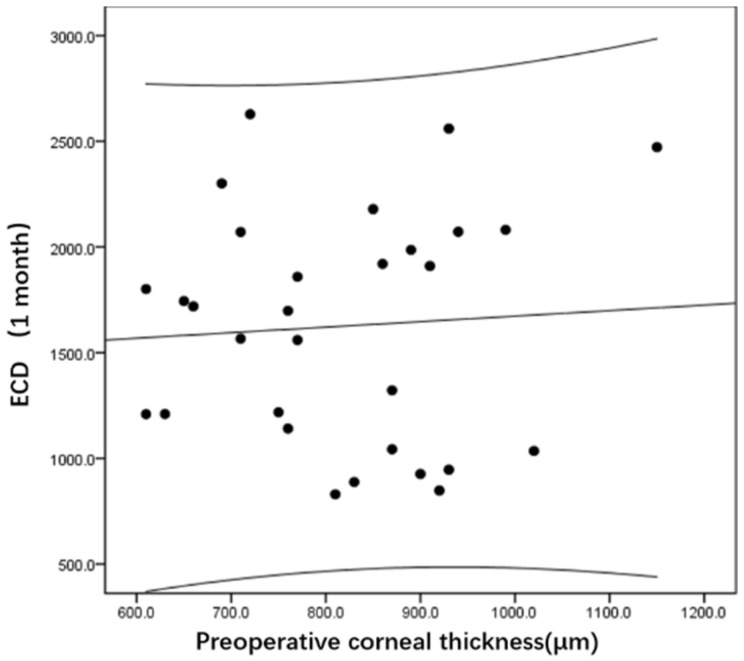
Relationship between the ECD after DSAEK (1 month) and the preoperative corneal thickness.

**Figure 4 jcm-11-05601-f004:**
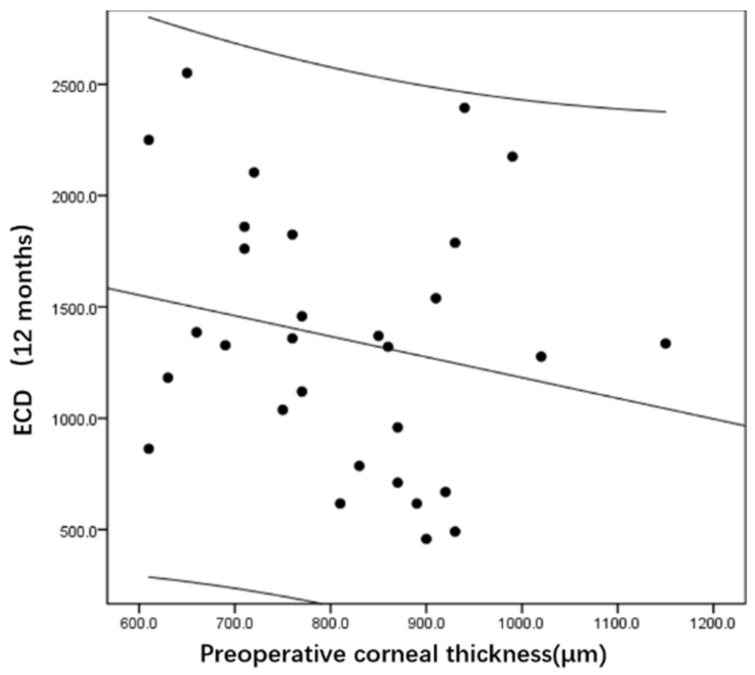
Relationship between the ECD after DSAEK (12 months) and the preoperative corneal thickness.

**Table 1 jcm-11-05601-t001:** Baseline characteristics of patients who underwent Descemet stripping automated endothelial keratoplasty.

	Group 1 (n = 15)	Group 2 (n = 15)	*p*-Value *
Males, n (%)	6 (40.0)	6 (40.0)	1.00
Age (years), mean ± SD	57.0 ± 16.1	59.5 ± 24.4	0.74
Eye OD, n (%)	8 (53.3)	12 (80)	0.12
Diagnosis, n (%)			0.55
PBK	12 (80.0)	9 (60.0)	
Fuchs endothelial dystrophy	2 (13.3)	2 (13.3)	
Trauma	1 (6.7)	2 (13.3)	
CHED	0 (0)	2 (13.3)	
Preoperative BCVA (logMAR)	1.21 ± 0.64	1.53 ± 0.70	0.16
Preoperative corneal thickness (μm)	707.6 ± 63.6 (range)	924.0 ± 80.7 (range)	<0.001
Triple-DSAEK, n (%)	3 (20.0)	3 (20.0)	1.0
Donor ECD (cells/mm^2^)	3445 ± 348	3222 ± 343	0.08

PBK = pseudophakic bullous keratopathy; CHED = congenital hereditary endothelial dystrophy; ECD = endothelial cell density; SD = standard deviation; DSAEK = Descemet stripping automated endothelial keratoplasty; Triple-DSAEK = DSAEK combined with phacoemulsification and intraocular lens implantation; * *t*-tests for continuous variables and chi-square tests for categoric variables.

**Table 2 jcm-11-05601-t002:** Outcomes after DSAEK in patients with different preoperative cornea thickness at baseline.

	Group 1 (n = 15) Mean ± SD	Group 2 (n = 15) Mean ± SD	*p*-Value *
BCVA (logMAR)			
1 month	0.79 ± 0.37	0.80 ± 0.30	0.59
12 months	0.53 ± 0.28	0.59 ± 0.25	0.28
ECD (cells/mm^2^)			
1 month	1637 ± 475	1613 ± 626	0.91
12 months	1513 ± 537	1193 ± 600	0.13
Recipient corneal thickness, RCT (μm)			
1 month	508 ± 52	592 ± 126	0.03
12 months	529 ± 53	539 ± 56	0.63
Graft thickness, GT (μm)			
1 month	117 ± 44	150 ± 59	0.08
12 months	109 ± 33	110 ± 54	0.95
Central cornea thickness (μm)			
1 month	625 ± 60	742 ± 143	0.008
12 months	638 ± 53	648 ± 75	0.806

ECD = endothelial cell density. * *t*-tests for continuous variables and chi-square tests for categoric variables.

## Data Availability

Results are contained within the article.

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
