# Peer review of "Descemet Stripping Automated Endothelial Keratoplasty in Thick Corneas"

_jcm, 2022, doi:10.3390/jcm11195601_

Round 1

Reviewer 1 Report

the authors wrote an interesting article. 

Multiple improvements are needed

Can you please specify better thick corneas? limits? how do you set the limits?

Please have a look and cite the following paper regarding Endothelial Transplants and DMEK as a potential substitution to DSAEK

PMID: 34306743 

PMID: 34768364 

PMID: 34447591

Please add a section of future studies and future directions

Please improve the English level

Please improve the limitations section

Author Response

Response to Reviewer 1 Comments

Dear reviewer, thank you for carefully and through reading our manuscript. The constructive comments and recommendations will help us to improve out manuscript. Our responses to several comments are listed below:

The authors wrote an interesting article. Multiple improvements are needed

Can you please specify better thick corneas? limits? how do you set the limits?

Response: The level of 820 micron was used as it was the median of the Corneal thickness measurements.

Please have a look and cite the following paper regarding Endothelial Transplants and DMEK as a potential substitution to DSAEK: PMID: 34306743, PMID: 34768364, PMID: 34447591

Response: These references have been added to the revised manuscript.

Please add a section of future studies and future directions

Response: The journal guidelines do not allow adding another section to the manuscript. We have added that, ‘Descemet’s membrane endothelial keratoplasty (DMEK) has been introduced as an alternative to DSAEK for quicker visual recovery. It would be interesting to compare DMEK and DSAEK in extremely thick recipient corneas in future studies.’

Please improve the English level

Response: We have revised the manuscript accordingly.

Please improve the limitations section

Response: We have revised to, ‘The main limitations for the current study are a small sample size and lack of a comparative surgical group. We did not have data from any additional follow-up visits. Long-term follow-up beyond 1-year postoperatively will help us to understand if grafts behave differently in thicker corneas. Descemet’s membrane endothelial keratoplasty (DMEK) has been introduced as an alternative to DSAEK for quicker visual recovery. It would be interesting to compare DMEK and DSAEK in extremely thick recipient corneas in future studies.’

Reviewer 2 Report

It is a well written article on the use of DSAEK in thick Corneas. The level of 820 micron was used as it was the median of the Corneal thickness measurements. A similar approach has been used by Holland et all in the past but it would make sense for the authors to elaborate further on this.

The indications for surgery as expected with these corneal thicknesses could be divided in Bullous Keratopathy and others and an analysis between the two could be done to greatly improve the paper.

It would make sense for the authors to elaborate why DSAEK as a
technique was used. Was it due to poor visualisation?

Also I have noticed that the graft thickness did not change much from the introduction to the 12 month period. It would be useful to have a comment on that. Was it because of the relative dehydration associated with dextran in the transport medium?

Overall an interesting read that would be greatly improved with some
more details.

Author Response

Response to Reviewer 2 Comments

Dear reviewer, thank you for carefully and through reading our manuscript. The constructive comments and recommendations will help us to improve out manuscript. Our responses to several comments are listed below:

It is a well written article on the use of DSAEK in thick Corneas. The level of 820 micron was used as it was the median of the Corneal thickness measurements. A similar approach has been used by Holland et all in the past but it would make sense for the authors to elaborate further on this. The indications for surgery as expected with these corneal thicknesses could be divided in Bullous Keratopathy and others and an analysis between the two could be done to greatly improve the paper.

Response: Thank you for the encouraging comments. We agree with the reviewer about further analysis. We further analyzed cases with and without bullous keratopathy. In cases with bullous keratopathy, compared to preoperative corneal thickness (795.2±108.9 microns), there was a significant change at 1 month (650.1±82.8 microns) and 12 months (641.2±62.9 microns). In non-bullous keratopathy cases, the preoperative corneal thickness (863.3±170.8 microns) did not change significantly at 1 month (762.3±167.6 microns) but the thickness at 12 months (648.3±71.1 microns) was significantly less compared to the preoperative level.

It would make sense for the authors to elaborate why DSAEK as a technique was used. Was it due to poor visualization?

Response: The choice for DSAEK was made due to suboptimal visualization, We have added this to the concluding paragraph of the revised manuscript.

Also, I have noticed that the graft thickness did not change much from the introduction to the 12 month period. It would be useful to have a comment on that. Was it because of the relative dehydration associated with dextran in the transport medium? Overall an interesting read that would be greatly improved with some more details.

Response: Thank you for the insightful comment. We agree that the graft thickness did not change much over the 12-month period. As the reviewer suggested, we hypothesize that it was likely due of the relative dehydration associated with dextran in the transport medium. Additionally, long-term follow-up might show a difference in graft thickness. We have added this comment to the last paragraph of the revised manuscript.

Round 2

Reviewer 1 Report

Authors improved the paper